# The Use of Teach Back at Hospital Discharge to Support Self-Management of Prescribed Medication for Secondary Prevention after Stroke—Findings from A Feasibility Study

**DOI:** 10.3390/healthcare11030391

**Published:** 2023-01-30

**Authors:** Sebastian Lindblom, Charlotte Ytterberg, Maria Flink, Axel C. Carlsson, Una Stenberg, Malin Tistad, Lena von Koch, Ann Charlotte Laska

**Affiliations:** 1Department of Neurobiology, Care Sciences and Society, Karolinska Institutet, 14183 Stockholm, Sweden; 2Theme of Women’s Health and Allied Health Professionals, Karolinska University Hospital, 14186 Stockholm, Sweden; 3Academic Primary Health Care Centre, Region Stockholm, 11365 Stockholm, Sweden; 4The Norwegian National Advisory Unit on Learning and Mastery in Health, Oslo University Hospital, 0372 Oslo, Norway; 5Frambu Centre for Rare Disorders, 1404 Siggerud, Norway; 6School of Health and Welfare, Dalarna University, 79131 Falun, Sweden; 7Theme of Heart & Vascular and Neuro, Karolinska University Hospital, 14186 Stockholm, Sweden; 8Department of Clinical Sciences Danderyd Hospital, Karolinska Institutet, 18288 Stockholm, Sweden

**Keywords:** health literacy, patient discharge, care transitions, rehabilitation, communication, medication adherence

## Abstract

The study aimed to investigate whether a structured discharge letter and the use of the person-centred communication method Teach Back for sharing information at hospital discharge could support perceived understanding and knowledge of and adherence to prescribed medication for secondary prevention after stroke. Data from a feasibility study of a codesigned care transition support for people with stroke was used. Patients who at discharge received both a structured discharge letter and participated in the person-centred communication method Teach Back (*n* = 17) were compared with patients receiving standard discharge procedures (*n* = 21). Questionnaires were used to compare the groups regarding perceived understanding of information about medical treatment, knowledge of information about medical treatment and medication adherence at 1 week and 3 months. There was a statistically significant difference in perceived understanding of information about medical treatment (*p* > 0.01) between the groups in favour of those who participated in Teach Back at the discharge encounter. No differences between groups were found regarding understanding health information about medical treatment and medication adherence. The results indicate that the use of Teach Back at the discharge encounter positively impacts perceived understanding of information about medical treatment in people with stroke. However, considering the nonrandomised study design and the small sample size, a large-scale trial is needed.

## 1. Introduction

The burden of stroke is expected to remain high globally, despite major advances in acute stroke medical management [1]. In Sweden, approximately 22,000 people with a mean age of 75 years have a stroke each year, of whom 20% have a recurrent stroke [2]. Secondary stroke prevention by medication is associated with reduced risk for a recurrent stroke and is highly recommended based on extensive evidence in both national [3] and international guidelines [4]. Nevertheless, nonadherence to prescribed secondary stroke prevention medication is common [5], and a rapid decline in adherence has been reported [6]. Secondary stroke prevention includes interventions to support health behaviours that reduce the risk of recurrent stroke, such as adherence to prescribed medication [7]. However, evidence-based guidelines on secondary stroke prevention that support health behaviours are scarce [8]. Perceived understanding of information provided at the time of hospital discharge has been identified as crucial for the continued use of prescribed secondary stroke prevention medication [9,10].

The importance of understanding information has been stressed by the WHO, which defines health literacy as the knowledge and competence that enables people to “access, understand, appraise and use information and services in ways that promote and maintain good health” [11]. In addition to the individual’s ability, health literacy also depends on the capability of healthcare organisations to provide services that support patients’ health literacy [12]. However, the short length of hospital stays for people with acute stroke in Sweden, a median of 7 days [2], places high demands on the effectiveness of stroke units to support patients’ health literacy. As low health literacy is associated with reduced understanding of and adherence to medical advice, greater healthcare utilisation and higher mortality [13], it has been strongly suggested that health literacy should be addressed to enhance, e.g., self-management of medication to promote health [14].

Our previous studies suggest that current hospital discharge services are not tailored to support patients’ health literacy and capacity for self-management postdischarge [15]. Information regarding own medication was the area that people with stroke were least satisfied with [16]. Other studies have identified that older people may be particularly vulnerable during the discharge phase; up to 62% could not name their new medications postdischarge [17], and more than half could not recall the follow-up appointments that were planned [18]. Moreover, despite 90% stating that they had understood the discharge information, 40% could not recall their diagnosis or recall the healthcare plan after hospital discharge [19]. Consequently, there is a gap between the information provided and the information patients understand and apply. Such gaps potentially have great risks for the individual patient. Of 1500 hospitalised patients, 11.9% were reported to have new or worsening symptoms within 3–5 days after hospital discharge [20]. Further, it has been reported that about one in five patients were affected by adverse drug events following discharge from hospital [21]. Thus, it is crucial that health care services are tailored to meet patients’ varying levels of health literacy to ensure patients’ understanding of the information to enable self-management postdischarge.

In our previous studies of the care transition from hospital with referral to a neurorehabilitation team in primary care, we identified a need for an improved dialogue between patients and healthcare professionals to support self-management after discharge [22]. Teach Back is a person-centred mode of iterative communication focusing on shared understanding of the patient’s situation and the professional’s information [23]. The use of Teach Back for sharing information has been shown to reduce the number of hospital readmissions [24] and improve medication adherence and self-management for people with chronic conditions [25], but it has not specifically been applied to people with stroke. Therefore, the aim of the present study was to investigate whether a structured discharge letter and the use of the person-centred communication method Teach Back for sharing information at hospital discharge could support perceived understanding and knowledge of and adherence to prescribed medication for secondary prevention after stroke.

## 2. Materials and Methods

The study was carried out in the context of a feasibility study of a codesigned care transition support for people with stroke [26]. The care transition support was a multicomponent intervention. Components in the intervention to support health literacy for self-management of prescribed medication for secondary stroke prevention included a structured discharge letter on the patient’s health condition, the prescribed secondary preventive medications and the use of Teach Back at the discharge encounter. Teach Back is a person-centred communication method used to ensure a common understanding of the patient´s situation and the healthcare professional’s information. The method is an iterative process in which healthcare professionals ask the patient to recall information in their own words to ensure the patient´s comprehension of the information provided. If necessary, the healthcare professional clarifies and tailors information to the patient’s needs and asks the patient to recall the information again. If necessary, the cycle is repeated until a shared understanding is reached [23].

### 2.1. Participants and Procedures

The recruitment took place between October 2021 and June 2022. Participants were consecutively recruited at a stroke unit and a geriatric ward at a university hospital, hospital A, and at a stroke unit at a regional hospital, hospital B, in Stockholm, Sweden. Eligibility for inclusion were all patients with stroke who were to be discharged from the stroke units and geriatric ward at hospitals A and B, with referral to a neurorehabilitation team in primary care. Referral-based transitions to neurorehabilitation teams are an established stroke care pathway in Region Stockholm and have previously been reported in detail [27]. Eligible patients received oral and written information about the study, and written informed consent was obtained. Patients were purposively allocated to the intervention or control group. Participants who received both the structured discharge letter and the person-centred communication method Teach Back were allocated to the intervention group. Those who did not were allocated to the control group. At hospital A, physicians at the stroke unit and at the geriatric ward were instructed to use the intervention at discharge. The comparison group received the regular discharge procedure from the participating hospital. Baseline data were collected through questionnaires and from medical records by research assistants at the hospitals. Questionnaires were mailed 1 week and 3 months after discharge to obtain follow-up data. A research assistant contacted the participants by telephone to assist in filling out the questionnaires, which were returned in a prepaid envelope. The study was approved by the Swedish Ethical Review Authority and registered at www.clinicaltrials.gov (accessed on 15 novemeber 2022) id: NCT02925871.

### 2.2. Data Collection

Baseline characteristics comprising age, sex, educational level (elementary/secondary or university/college), civil status (living alone or cohabiting), work status (working or not working) and information on use of home care services before stroke (yes or no) was collected using a questionnaire. Health literacy was assessed with the Health Literacy Questionnaire [10], comprising nine scales that each measure an aspect of health literacy. Subscale 2, “Having sufficient information to manage my health”, which is comprised of four questions, and subscale 9, “Understand health information well enough to know what to do”, which is comprised of five questions, were used in the present study. The respondent was asked to answer each question on a Likert scale ranging from 1 (strongly disagree or cannot do) to 5 (strongly agree or very easy). The mean score across all questions was calculated for each subscale.

Disease-related data comprised length of hospital stay, type of stroke (ischaemic stroke or intracerebral haemorrhage), reperfusion therapy (yes or no), stroke severity assessed using the National Institute of Health Stroke Scale [28], aphasia and comorbidity were obtained from patient records. The Charlson Comorbidity Index [29] was used to categorise comorbidity into 3 grades of severity: no comorbidity (scores 0), low (scores 1 and 2) and moderate or severe (scores > 2). The modified Rankin Scale [30], with scores ranging from 0 (no disability) to 6 (death), was used to assess the degree of disability, categorised as mild (0–1), moderate (2–3) and severe (4–6) disability. The short version of the Montreal Cognitive Assessment Scale was used to assess cognitive function [31,32]. The scores range from 1 to 15, and scores < 11 indicate cognitive impairment. The Patient Health Questionnaire-2 [33] was used to assess depression symptoms. The total score ranges from 0 to 6, and a score of >3 is considered to indicate depressive symptoms [31]. The Barthel Index (BI) was used to assess activities of daily living (ADL) [34]. The BI assesses independence in ten self-care and mobility activities, and the total score ranges from 0 to 100, where a higher score reflects a higher degree of independence. Walking ability was categorised as walks independently without aid and support, walks with walking aid or unable to walk/walks with assistance and support. Perceived recovery was rated by the participants on a subscale of the Stroke Impact Scale 3.0, a visual analogue scale from 0 (maximum perceived impact) to 100 (no perceived impact) [35,36].

Perceived understanding of information about medical treatment at one week was assessed using the Swedish version of the Care Transition Measure (CTM-15) [37], items 13–15. The CTM-15 contains 15 items that assess perceived quality in care transitions, in which items 13–15 (CTM-Medication) concern understanding the purpose of taking medications, how to take them and possible side effects. The respondent was asked to agree or disagree with each statement on a Likert scale ranging from 1 (strongly disagree) to 4 (strongly agree). For each item, there is also an additional response alternative of don’t know/not applicable. The mean and median for the total score of CTM-Medication and each item were calculated. Knowledge of information about medical treatment was also assessed using open-ended questions concerning knowledge of which new medications had been prescribed at the time of discharge from hospital and/or changes in previous medications, as well as knowledge of the reasons for new medications or changes in medication. Responses were verified against medical records and dichotomised into has knowledge or has no knowledge.

Adherence to medical treatment was assessed using the Swedish version of the Medication Adherence Report Scale (MARS-5) [38]. The MARS-5 is a self-report scale containing five items assessing nonadherent behaviour, both intentional (“I change the dosage of my medications”) and nonintentional (“I forget to take my medications”). The respondent was asked to agree or disagree with each statement on a Likert scale ranging from 1 (always) to 5 (never). The total score ranges from 5 to 25, where higher scores indicate higher adherence.

### 2.3. Analyses

Descriptive statistics were used to present the participants’ characteristics. To analyse differences between participants who received the intervention and those who did not, the Mann–Whitney U test or the Student’s *t*-test was used for continuous data and the Fisher’s Exact Test for categorical data. The level of significance was set at *p* ≤ 0.05.

## 3. Results

A total of 44 participants were included in the study, whereof 28 were from hospital A and 16 from hospital B. Of these, 19 participants received the intervention, and 25 did not. A total of six participants declined further participation at the 1-week follow-up and were not included in the final analysis. Additionally, four participants declined further participation at the 3-month follow-up; see the flow diagram in Figure 1. The median age of the 38 participants included in the analysis was 77 years (IQR 75–81); 55% were men, and 66% had a mild stroke. The participants lost to follow-up had a median age of 77 years (IQR 73–81), 50% were men and 50% had a mild stroke.

The demographic and clinical characteristics of the participants are presented in Table 1. The baseline characteristics of the intervention and control groups were comparable except for the level of education, as 60% in the intervention group had a university degree in comparison to 29% in the control group (*p* = 0.049).

The between-group analyses between those who received the intervention and those who did not showed a statistically significant difference in the total score of perceived understanding of information about medical treatment (*p* > 0.01) in favour of the intervention group, see Table 2.

Participants who received the intervention reported a higher perceived understanding of the purpose of taking their medications (*p* = 0.024) and how to take them (*p* = 0.01) than participants who did not receive the intervention. There was no difference in perceived understanding of possible side effects of their medications between the groups (*p* = 0.065). There were also no differences in knowledge of new medications or of changes in medication, see Table 3.

Regarding adherence to medical treatment, presented in Table 4, no differences between groups were found.

## 4. Discussion

This study aimed to investigate whether using Teach Back and a structured discharge letter for sharing information at the time of hospital discharge could support patient knowledge of and adherence to prescribed medication.

To our knowledge, this is the first study to investigate the use of Teach Back in people with stroke. Although no differences were found regarding adherence to medical treatment, the findings revealed that participants who received the person-centred communication method Teach Back at the discharge encounter had a higher perceived understanding of information about medical treatment 1-week postdischarge than those who received the standard discharge procedure.

Our results indicate that the use of Teach Back at the discharge encounter improved the perceived understanding of information about medical treatment measured using the CTM-Medication. Our results are in line with those of previous studies in other populations showing that patients perceive Teach Back to be an effective educational method [39] and that the use of Teach Back is associated with improved communication with healthcare professionals [40], greater patient understanding and empowerment [41] and improved self-efficacy [42].

There were no significant differences between the groups with regard to the knowledge of new medications or changes in medication or medication adherence at 1 week or 3 months. Previous studies have shown that Teach Back can have a positive effect on knowledge retention in hospitalised heart failure patients [43] and medication adherence in patients with diabetes [44]. However, both studies provided a more comprehensive educational programme, spanning over a more extended time period than was the case in our study, and found that the greater time spent in education was associated with better knowledge retention [43]. Furthermore, person-centred communication, including repeated personalised information, has been shown to improve medication adherence [45]. The intervention in the present study was only provided on one occasion, which might not have been enough to support the self-management of prescribed medication.

The intervention and control groups were similar with regard to baseline values, except for education level. As a higher level of education has been shown to be associated with higher health literacy [46], the larger proportion of participants with a university degree in the intervention group may have impacted the results. However, the level of health literacy was similar between the groups, which indicates that the intervention was successful in terms of perceived understanding of information about medical treatment. Further, even though there was no significant difference, the control group had a higher number of people cohabiting. This might have affected medication adherence results, as family support helps create routines that aid people in taking their medication [47].

No differences were seen with regard to medication adherence. In the present study, we measured medication adherence using MARS-5. A majority of participants reported total scores on MARS-5, indicating a ceiling effect of the instrument. This is something that has also been reported elsewhere [48]. Furthermore, the use of self-reported measures of adherence is problematic, as people often fail to remember the instructions or whether they have taken their medication according to instructions or not, and they also tend to overreport their adherence [49].

### Limitations 

The findings of the present study should be interpreted with caution due to the nonrandomised study design and the small sample size. Further, a majority of the participants had a mild stroke, which limits the extrapolation of our findings to the general stroke population.

## 5. Conclusions

The results from this study indicate that the use of the person-centred communication method Teach Back in the discharge encounter has a positive impact on the perceived understanding of information about medical treatment 1-week postdischarge in people with stroke. However, there is a need for a large-scale trial to determine the effect of Teach Back on knowledge of information to support self-management of prescribed medication for secondary prevention after stroke.

## Figures and Tables

**Figure 1 healthcare-11-00391-f001:**
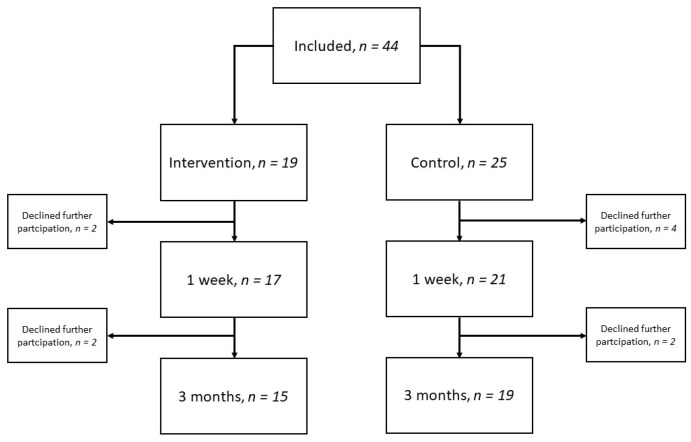
Flow diagram over included participants and time-point for follow-up.

**Table 1 healthcare-11-00391-t001:** Demographics and clinical characteristics of the participants.

Variable	Value			
	Total *n* = 38	Intervention *n* = 17	Control *n* = 21	*p*-Value
Age, median (IQR)	75 (69–82) 27–94	75 (64–85) 27–91	75 (69–80) 53–94	0.724 ^a^
Sex, male, *n* (%)	21 (55)	9 (53)	12 (57)	
Education, *n* (%)				0.049 ^b^
Elementary/secondary	22 (58)	6 (35)	15 (71)	
University	16 (42)	10 (60)	6 (29)	
Cohabiting, *n* (%)	24 (63)	9 (53)	15 (71)	0.385 ^b^
Working, *n* (%)	10 (26)	6 (35)	4 (19)	0.293 ^b^
Home care services before stroke, *n* (%)	2 (5)	1 (6)	1 (5)	0.701 ^b^
Health literacy				
Subscale 2. Having sufficient information to manage my health, mean (SD) 95% CI	2.9 (0.6) 2.7–3.1	3 (0.7) 2.6–3.3	2.9 (0.6) 2.6–3.1	0.597 ^c^
Subscale 9. Understand health information enough to know what to do, mean (SD) 95% CI	3.9 (0.9) 3.6–4.2	4 (0.9) 3.5–4.5	3.9 (0.8) 3.5–4.2	0.538 ^c^
Length of stay, days, median (IQR) min–max	3 (2–5) 1–15	3 (2–7.5) 1–12	3 (2–4) 1–15	0.634 ^a^
Type of stroke, *n* (%)				1.0 ^b^
Ischaemic	35 (92)	16 (94)	19 (90)	
Intracerebral haemorrhage	3 (8)	1 (6)	2 (10)	
Reperfusion therapy, *n* (%)	3 (8)	2 (12)	1 (5)	0.577 ^b^
NIHSS	1.5 (1–3) 0–18	2 (2–7.5) 1–12	1 (0–3) 0–13	0.683 ^a^
Aphasia, *n* (%)	3 (8)	1 (6)	2 (10)	1.0 ^b^
Comorbidity, *n* (%)				0.424 ^b^
No comorbidity	15 (39)	8 (47)	7 (33)	
Low comorbidity	20 (53)	7 (41)	13 (62)	
Moderate/severe comorbidity	3 (8)	2 (12)	1 (5)	
Disability, *n* (%)				0.523 ^b^
Mild	25 (66)	12 (71)	13 (62)	
Moderate	12 (32)	5 (29)	7 (33)	
Severe	1 (2)	-	1 (5)	
MoCA, signs of impaired cognition, *n* (%)	18 (47)	11 (65)	7 (33)	0.101 ^b^
PHQ-2, signs of depression, *n* (%)	10 (26)	3 (18)	7 (33)	0.281 ^b^
Barthel Index, median (IQR) min–max	100 (100–100) 65–100	100 (100–100) 65–100	100 (98–100) 80–100	0.650 ^a^
Walking ability, *n* (%)				0.684 ^b^
Walks independently without aid and support	29 (76)	13 (77)	16 (76)	
Walks with walking aid	8 (21)	3 (18)	5 (24)	
Unable to walk/walks with assistance and support	1 (3)	1 (6)	-	
SIS recovery, median (IQR) min–max	75 (60–90) 20–100	75 (63–84) 50–100	70 (55–90) 20–99	0.868 ^a^

Abbreviations: IQR, interquartile range; NIHSS, National Institute of Health Stroke Scale; MoCA, Montreal Cognitive Assessment; PHQ-2, Patient Health Questionnaire-2; SIS, Stroke Impact Scale. ^a^ Mann–Whitney U test, ^b^ Fisher exact test, ^c^ Student’s *t*-test.

**Table 2 healthcare-11-00391-t002:** Comparison between intervention group and control group for perceived understanding of information about medical treatment 1-week postdischarge.

	Total, *n* = 38	Intervention, *n* = 17	Control, *n* = 21	*p*-Value *
CTM-3 Total				
Mean, SD	72 (29)	83 (26)	62.6 (29)	
Median, IQR	72 (56–100)	100 (72–100)	66.7 (56–89)	0.01
CTM item 13, I clearly understood the purpose of taking each of my medications				
Mean, SD	3.4 (0.9)	3.7 (0.8)	3.1 (0.9)	
Median, IQR	4 (3–4)	4 (3.5–4)	3 (3–4)	0.024
CTM item 14, I clearly understood how to take each of my medications, including how much I should take and when				
Mean, SD	3.4 (0.9)	3.7 (0.8)	3.1 (0.9)	
Median, IQR	4 (3–4)	4 (4–4)	3 (3–4)	0.01
CTM item 15, I clearly understood the possible side effects of each of my medications				
Mean, SD	2.8 (1.1)	3.1 (1.2)	2.4 (1)	
Median, IQR	2.5 (2–4)	4 (2–4)	2 (2–3)	0.065

* Mann–Whitney U test.

**Table 3 healthcare-11-00391-t003:** Comparison between intervention group and control group for verified knowledge of new medications or changes in medication at 1 week and 3 months.

	Total		Intervention		Control		*p*-Value *
1 Week	Yes	No	Yes	No	Yes	No	
Knowledge of new medication, *n* (%)	30 (81)	7 (19)	14 (82)	3 (18)	16 (80)	4 (20)	1.0
Knowledge of changes in medication, *n* (%)	29 (78)	8 (22)	14 (92)	3 (18)	15 (75)	5 (25)	0.701
**3 months**							
Knowledge of new medication, *n* (%)	18 (53)	16 (47)	8 (53)	7 (47)	10 (53)	9 (47)	1.0
Knowledge of changes in medication, *n* (%)	27 (82)	6 (18)	13 (87)	2 (13)	14 (78)	4 (22)	0.665

* Fisher exact test.

**Table 4 healthcare-11-00391-t004:** Comparison between intervention group and control group for medication adherence at 1 week and 3 months.

	Total	Intervention	Control	*p*-Value *
MARS-5 Total				
**1 week**				
Mean	24 (3.4)	24.2 (1.5)	23.9 (4.4)	0.293
Median	25 (25–25)	25 (24–25)	25 (25–25)	
Min–Max	5–25	20–25	5–25	
**3 months**				
MARS-5 Total				
Mean	24.3 (2.1)	24.4 (0.9)	24.1 (2.8)	0.289
Median (IQR)	25 (24–25)	25 24–25	25 (25–25)	
Min–Max	13–25	22–25	13–25	

* Mann–Whitney U test.

## Data Availability

The datasets generated and/or analysed during the current study are not publicly available but can be available upon reasonable request. As data can indirectly be traced back to the study participants, according to the Swedish and EU personal data sharing legislation, access can only be granted upon request. Request for access to the data can be put to our Research Data Office (rdo@ki.se) at Karolinska Institutet and will be handled according to the relevant legislation. In most cases, this will require a data processing agreement or similar with the recipient of the data.

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
