# Peer review of "The Use of Teach Back at Hospital Discharge to Support Self-Management of Prescribed Medication for Secondary Prevention after Stroke—Findings from A Feasibility Study"

_healthcare, 2023, doi:10.3390/healthcare11030391_

Round 1

Reviewer 1 Report

Thank you for asking me to view this article.

The topic under study is very interesting especially in relation to the fact that the implementation of teaching and learning strategies such as "Teach-back" seem to represent a useful tool not only in relation to the patient's therapeutic adherence but also for that concerning the quality of care. However, despite the topicality of the topic explored by the authors, Major revision is considered necessary before proceeding with a further revision process. Here are my suggestions in this regard.

The abstract should briefly summarize the context in which the research question is developed, the methodology chosen to answer the question, the results obtained and a brief conclusion. The description of the contents, therefore, should faithfully summarize the sections of which the manuscript is composed. What the reader deduces from the abstract is not entirely consistent with what is detailed in the introduction and methods. For example, knowledge of the "Teach Back" strategy is taken for granted by the reader, therefore it would be useful to insert its definition or at least a reference that clarifies its meaning. Furthermore, in the text the authors speak of a questionnaire while in the abstract there is no mention of the methodology chosen to investigate the usefulness of the strategy in terms of health literacy and patient self-management. It is suggested to review the abstract and organize its contents so that they are consistent with the text.

In my opinion, the introductory section is described in a dispersive and unordered way. This diverts attention from the study's objective of investigating whether the use of "Teach Back" for sharing information at hospital discharge could support self-management of prescribed secondary prevention medications after stroke.

For the methods section, I suggest the authors insert a paragraph on the "study setting" that best describes the context in which the study is developed. For example, in line 102 reference is made to a "hospital A" without mentioning its characteristics which instead are described only in the following paragraph (lines 107-111) in which hospitals A and B are identified. Furthermore, I suggest that the authors better describe the method of recruitment and the type of intervention should be detailed. Having chosen to administer a questionnaire together with telephone interviews, it would be useful to also describe the time dedicated to each patient during the interviews as it is well known in the literature that health education interventions positively amplify the possibility of obtaining more effective results when accompanied by dedicated meetings (doi: 10.3390/ijerph191811212). Moreover, for the purpose of reproducibility of the study also in other contexts, it would be useful to insert the questionnaire as it was administered as supplementary material.

Reviewer 2 Report

I commend the authors for addressing a great need in post stroke care. 

1. Can the authors comment if patients with aphasia/ language disorder were excluded from the study? How is the CTM 15 used in this population.  

2. How were patients selected. Were these consecutive patient or were people with moderate to severe stroke excluded. For the MARS -5 what if patients were in supervised settings that gave them medication thus taking responsibility away from the patient. Just trying to clarify whom the patient study population is and whom optimally benefits from this intervention. 

3.  Can the authors comment on the disparity between university education being higher in the intervention group. Given the small numbers of study paitents could this not skew results?

4. Was socioeconomic status included?

5. Can you comment on the incerased number of cohabiting in control - should that not push for more adherence?

Reviewer 3 Report

Dear Authors,

Thank you for the opportunity to review this very interesting and clinically useful work.

I am asking you to consider some suggestions:

1. Title - please consider whether, due to the limited size of the study and control sample, it is not better to add the annotation "preliminary study" in the title.

2. Introduction - it is worth considering changing the literature so that it is from the last 10 years

3. Materials and methods - it would be good to move the information on patient recruitment and flow-chart from the result section to the materials and methods section

4. Results - in the future, it would be good to increase the size of the study group in order to analyze socio-economic and biological factors that may also have a significant impact on the researched issue.

5. Please add the limitations section and describe them there. 

6. References - I kindly ask you to edit the references in accordance with the requirements of the journal. 

Kinds regards,
Reviewer

Reviewer 4 Report

Manuscript ID: healthcare-2123405-peer-review-v1

Manuscript title: The use of Teach Back at hospital discharge to support self-management of prescribed medication for secondary prevention after stroke

Comments

This manuscript reports a study aimed to investigate if the use of “Teach Back” for sharing information at hospital discharge could support self-management of prescribed medication for secondary prevention after stroke. The manuscript is generally well-written and organized. I have a few comments for the authors to consider before a final decision.

Major comments

1. The Introduction section poses a clear background and rationale for the study. The study aim could be rephrased, e.g. as posed in the abstract.

2. Materials and Methods. I missed some reporting guidelines to make sure important information is reported. Consider visiting the EQUATOR Network (https://www.equator-network.org) and searching for reporting guidelines related to a feasibility study design (e.g., https://www.equator-network.org/reporting-guidelines/consort-2010-statement-extension-to-randomised-pilot-and-feasibility-trials/).

3. Material and Methods, lines 96-104. I missed an explanation about group allocation. I assume this was not a randomized trial as per the abstract description, but any allocation strategy (purposive included) should be reported.

4. Results, lines 183-187. Consider reporting how many participants from each hospital (A and B) were included.

5. Results, lines 183-187. The dropout rates seem high, ~21% and ~24% for intervention and control groups respectively. A sensitivity analysis exploring differences in demographic variables between included and excluded participants may help to identify factors associated with dropout.

Minor comments

1. Abstract, line 23. Consider using ‘usual care’ instead of ‘standard discharge procedure’.

2. Abstract, lines 31-33. Consider suggesting further large randomized controlled trials instead of reporting the study’s design limitations (as in the Conclusions of the main text).

Round 2

Reviewer 1 Report

Thanks for responding to my comments. The authors responded comprehensively to the comments raised.

Reviewer 4 Report

Thank you for providing a letter of response to my previous comments. All comments were adequately addressed. I have no new comments.